# Evaluating the impact of a common elements-based intervention to improve maternal psychological well-being and mother–infant interaction in rural Pakistan: study protocol for a randomised controlled trial

Zill-e- Huma [1,2,3] Ayella Gillani,[1,3] Fakhira Shafique,[1,3] Alina Rashid,[1,3] Bushra Mahjabeen,[1,3] Hashim Javed,[1,3] Duolao Wang [4] Atif Rahman,[2] Syed Usman Hamdani[1,2,3]

For numbered affiliations see end of article.

**Correspondence to**
Zill-e- Huma;
zille.huma@hdrfoundation.org

## ABSTRACT

**Introduction** Millions of children in low resource settings are at high risk of poor development due to factors such as under nutrition, inadequate stimulation and maternal depression. Evidence-based interventions to address these risk factors exist, but often as a separate and overlapping package. The current study aims to evaluate the effectiveness of a common elements-based intervention to improve mother–infant interaction at 12 months post-partum.

**Method and analysis** A two-arm, single-blinded, individual randomised controlled trial is being carried out in the community settings of the rural subdistrict of Gujar Khan in Rawalpindi, Pakistan. 250 pregnant women in third trimester with distress (Self-Reporting Questionnaire, cut-off score >9) have been randomised on 1:1 allocation ratio into intervention (n=125) and treatment-as-usual arms (n=125). The participants in the intervention arm will receive 15 individual sessions of intervention on a monthly basis by non-specialist facilitators. The intervention involves components of early stimulation, learning through play, responsive feeding, guided discovery using pictures, behavioural activation and problem solving. The primary outcome is caregiver–infant interaction at 12 months postpartum. The secondary outcomes include maternal psychological well-being, quality of life, social support and empowerment. Infant secondary outcomes include growth, nutrition and development. The data will be collected at baseline, 6 and 12 months postpartum. A qualitative process evaluation will be conducted to inform the feasibility of intervention delivery.

**Ethics** Ethics approval for the present study was obtained from the Human Development Research Foundation Institutional Review Board, Islamabad Pakistan.

**Dissemination** If proven effective, the study will contribute to scale-up care for maternal and child mental health in low resource settings, globally. The findings of the present study will be published in peer-reviewed journals and presented at conferences and community forums.

**Trial registration number** NCT04252807.

## Strength and limitations of this study

► To address the burden associated with poor infant development and perinatal distress, we have developed a common elements-based early intervention. The intervention is designed to be delivered through non-specialist facilitators in low resource community settings.

► In a single-blinded, individual randomised controlled trial, we will evaluate the impact of intervention using a pragmatic primary outcome measure that is, 5 min observation of a live caregiver–child interaction.

► The COVID-19 pandemic has resulted in limited face-to-face contact with the study participants and the intervention is being delivered via telephone or in-person sessions where possible. We shall evaluate the feasibility and acceptability of blended intervention delivery methodology using mixed-methods process evaluation.

► Continued participation of mother–infant dyads through the completion of the intervention and to the final assessment point at 12 months postpartum could be a challenge in the COVID-19 pandemic.

## INTRODUCTION

In South Asia, 35%–45% children are undernourished[1] and over 40 million lag behind in their cognitive and socioemotional development.[2] The situation is compounded by maternal depression, which affects over a quarter of all women.[3] Suboptimal functioning and agency in depressed mothers is a barrier to implementation of any intervention directed at the young child.[4] Together, these factors have a negative impact on provision of nurturing care that is essential for optimal development.[5–8]

Pakistan has a population of about 200 million people, of which three quarters live in rural areas. 47% of children under 5 years of age suffer from stunted growth and development in Pakistan.[9] By conservative estimate about eight million children do not achieve their full cognitive potential.[2] Prevalence studies in the proposed study area have reported high rates of perinatal depression in mothers ranging from 25% to 37% and a strong and independent association of maternal depression with low birth weight and under nutrition.[10] Studies have shown that antenatal stress can have detrimental effects on fetal development.[11] Lack of early interventions mean that many of these children might develop cognitive, social and emotional difficulties, contributing further to emotional and financial stress in the families.[12] Especially in the context of ongoing COVID-19 pandemic, which is indirectly impacting maternal and child health outcomes due to disruption of healthcare services and lack of access to existing maternal and child healthcare during lockdown, delivering early interventions in community settings becomes imperative.[13]

Considerable effort has gone into child survival programmes, leading to a reduction in child mortality rates over the last decade. However, improved child survival means there are many more children who are at-risk of developmental delay. Evidence-based interventions to address risk factors for poor child development exist, but often as separate and overlapping packages of interventions.

To address these challenges, we have developed a package of intervention using a Common Elements Approach (CEA)[14] that combines evidence-based elements from packages of care addressing early infant stimulation, responsive feeding and maternal distress.[15–17] These approaches recognise the importance of early critical years and role of the primary caregiver—normally the mother to provide nutrition and an optimal environment for interaction and stimulation. Common Elements Approach (CEA) is the concept of identifying components that overlap across multiple evidence-based practices for a specific problem area.[14] CEA combines a number of overlapping theoretical approaches into a single intervention with synergistic effects across the range of domains. The mother, in turn, receives psychosocial support, education and skills training specific to early child development (ECD), provided via a well-monitored and supervised delivery system.

We aim to evaluate the effectiveness of common elements-based intervention compared with usual care to improve mother–infant interaction along with improvements in maternal psychological well-being, infant growth, nutrition and development at 12 months postpartum in low resource rural community settings of Pakistan.

## HYPOTHESES
We hypothesise that common element-based integrated intervention will result in improving caregiver–child interaction measured using Observation of Mother and Child Interaction (OMCI) in intervention arm compared with the treatment-as-usual (TAU) in control arm, at 12 months' postpartum. Our secondary hypotheses (exploratory) are that our intervention will result in improvement in infant growth, nutrition and development and will improve maternal psychological well-being and quality of life.

## METHODS AND ANALYSIS
### Study design
A two-arm, single-blinded, individual randomised controlled trial with mixed-methods process monitoring and evaluation is being carried out to evaluate the effectiveness of intervention. The participants are recruited from a rural subdistrict, randomised into intervention and control arms, on a 1:1 allocation ratio.

### Patient and public involvement
The multicomponent intervention is based on existing community programmes developed by the research team working collaboratively with the people from local communities including mothers with postnatal depression, fathers, extended family members, Community Health Workers (CHWs) and local volunteers.[4 6 12 18] The relevant stakeholders (community representatives and gate keepers and mental health experts) were involved through consultative workshops (Theory of Change workshops) in the formative research to develop a hypothesised pathway for the implementation of intervention in the study subdistrict. The findings of the present study will be disseminated to participants through presentation at community forums.

### Study settings
The present study is being conducted in subdistrict of Gujar Khan in Rawalpindi Pakistan. The subdistrict represents a typical rural area of Pakistan and have poor demographic and health profile indicators (with high fertility rate—four births per woman and low female literacy rate, ie 45%),[19] where one in every four women suffer from perinatal depression.[19 20] The infant mortality rate in Rawalpindi has been estimated to be 55 per thousand live births, the under-five mortality rate is 82 per thousand live births and the percentage of underweight children below 5 years of age is 25%.[21] Primary healthcare is delivered through a network of Basic Health Units (BHUs), each providing care to about 15 000–20 000 people. Each BHU is staffed by a doctor, midwife, vaccinator and 15–20 village-based CHWs also called Lady Health Workers (LHWs). These women have completed secondary school and are trained to provide preventive maternal and child healthcare, and education in the community. Each LHW is responsible for about 100 households in her village. The participants shall be recruited from the catchment areas of LHWs. The study participants are permanent residents in the study subdistrict and have been living here for past many

generations. The implementing research organisation has been working closely with the health department in the study area for last 15 years and has developed a trustable relationship with the community members. The study participants are being approached through LHWs, who have trusting relationship with community, this makes the enrolment process and retention in the study more feasible. Also, convenience of participants is taken into account when assessment and intervention delivery sessions are scheduled. These factors are expected to improve the retention rate of participants in study over a longer period of time.

## Research participants
The target population for the present study is pregnant women in third trimester with distress (screen positive on Self-Reporting Questionnaire (SRQ), cut-off score ≥9).

### Eligibility criteria of participants
#### Inclusion criteria
► Pregnant with third trimester (28 gestational week)
► Age 18–40 years
► Intend to reside in the study areas until the completion of the study
► Score ≥9 on the SRQ

#### Exclusion criteria
► Women who require immediate or on-going medical or psychiatric care reported.
► Severe previous or current obstetric morbidity including eclampsia and antepartum haemorrhage.
► Medical disorders that require inpatient management (eg, diabetes, hypertension, thromboembolism cardiac disease).

### Sample size calculations
The power calculations are based on one primary outcome, that is, mother–child interaction on OMCI tool[22] at 12 months' postpartum. The scoring of tool is based on the frequency of the occurrence of responsive behaviours of mother with higher scores indicating more responsive interactions. The effect size for the present study was estimated based on the findings of a similar study conducted in Pakistan, where OMCI was used as one of the outcomes to evaluate the effect of responsive stimulation intervention on caregiver–child interaction at 12 and 24 months postpartum.[8] The findings of the study showed mother–child dyads receiving responsive caregiving intervention had significantly higher mean scores on OMCI at 12 months (mean 32.3 (SD: 8.3) vs mean 27.1 (SD: 8.2), p<0.0001) than those who were allocated to control arm. Moreover, repeated-measures analysis showed that effects were sustained at 24 months. As the effect sizes for OMCI in Yousafzai *et al* 2015's study ranged between 0.6 and 0.9; therefore, a conservative effect size of 0.4 (two-sides hypothesis) at 12 months post-partum was proposed in the present study. The effect size was calculated using the two-sided t-test. Assuming an effect size of 0.4, with 80% power, 0.05 significance and a two-sided hypothesis test and accounting for 20% attrition, we will need 250 caregiver–child dyads (125 in each arm).

## Recruitment procedure
Before recruiting participants in the trial, the local community stakeholders were taken on-board. As mentioned in the study settings, the current study is being conducted in the rural subdistrict of Gujar Khan, Rawalpindi district, Pakistan. The study subdistrict consists of 36 Union Councils (UCs)—the smallest administrative units of the subdistrict. Each UC has a Basic Health Unit (BHU) that provides primary healthcare to about 15 000–20 000 local population. Each BHU is staffed by a doctor, midwife, vaccinator and 15–20 village-based CHWs also called LHWs. These women have completed secondary school and are trained to provide preventive maternal and child healthcare, and education in the community. Each LHW is responsible for about 100–150 households in her village and keeps a register of every new pregnancy in her catchment area. The research participants in the present study are being recruited from the catchment areas of LHWs, where, every woman in their third trimester of pregnancy is being assessed for eligibility. Pregnant women in their third trimester (n=250), who fulfil the eligibility criteria, are being enrolled in the trial and individually randomised into intervention and TAU arm (1:1 ratio) (figure 1). Written informed consent is obtained from the study participants by trained assessment team. A complete record of assessment against eligibility criteria, and reasons for refusal is maintained for those who do not consent for participation in the trial.

All potential eligible pregnant women in third trimester are screened for psychological distress by research team using SRQ. Psychological distress is represented by subscales of depression/anxiety, somatic symptoms, reduced vital energy and depressive thoughts during the past month. The scale reflects multidimensional nature of mental illnesses. It has 20 items scored on 0 or 1. A score of 1 indicates that the symptom was present during the past month; a score of 0 indicates that the symptom was absent. The maximum score is therefore 20, with a higher score indicating higher psychological distress. A score of ≥9 indicates increased risk of postnatal depression[8] and mothers with these scores have been recruited in the study.

## Intervention
The intervention called 'Nurturing Care' combines common elements from evidence-based interventions for maternal psychosocial distress (eg, WHO Thinking Healthy Programme[17]), early stimulation and responsive feeding (eg, Unicef Care for Child Development Package[15]) and child care (eg, WHO Caregivers Skills Training programme[16]) for optimal development of at risk mother and children.

The intervention package aims to improve (a) mother psychological distress by promoting problem solving, developing empathic listening, improving family support,

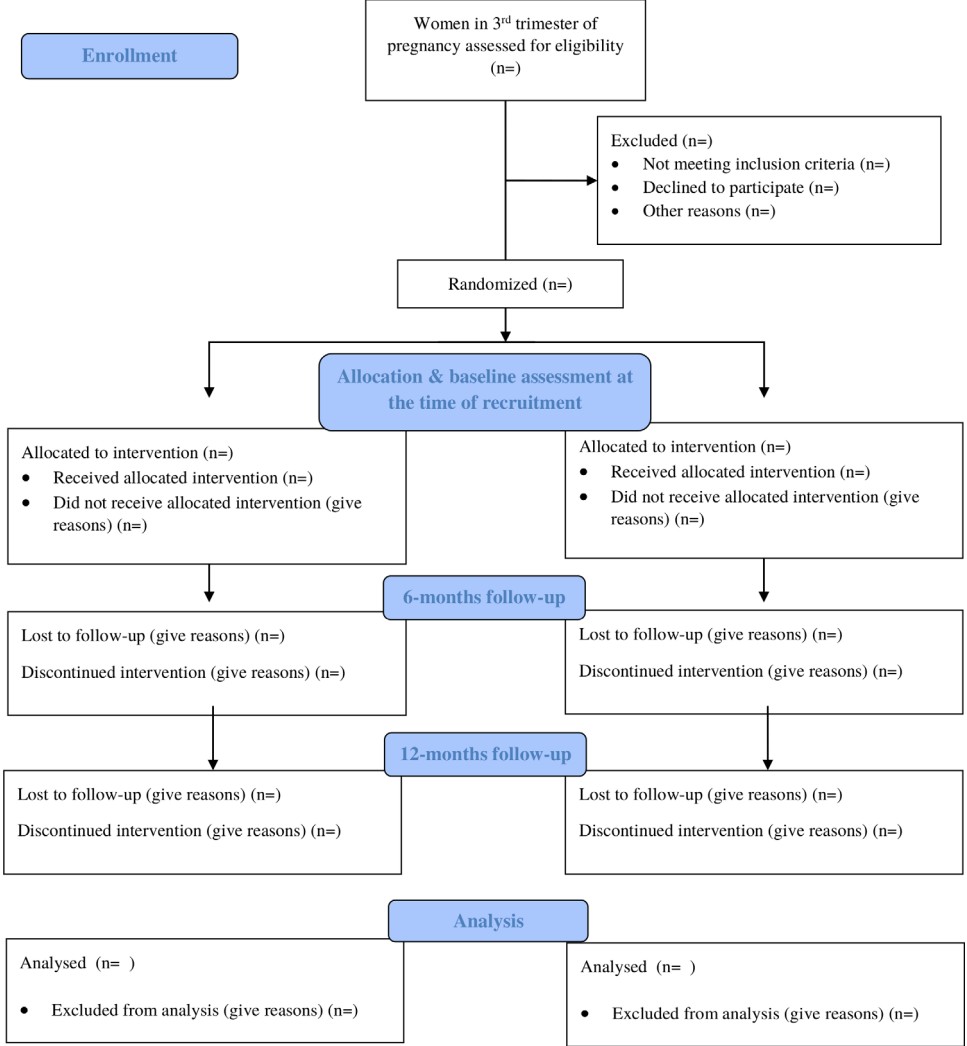

**Figure 1** Flow diagram.

using guided discovery through pictures and behavioural activation; (b) child development through psycho-education, responsive breastfeeding and (c) mother–infant interaction through early stimulation and learning through play (see table 1 for description of intervention sessions).

The intervention is designed to be delivered by non-specialist facilitators in 15 monthly sessions in community settings and commences from third trimester of pregnancy and ends at 12 months postpartum. During the COVID-19 pandemic, when the in-person sessions are not possible, we are delivering the scheduled intervention session via telephone. All those sessions delivered via telephone are not repeated in-person.

### Training and supervision of non-specialist facilitators in 'Nurturing Care' intervention

The delivery agents of the 'Nurturing Care' intervention are female non-specialist facilitators who are graduates in psychology/social sciences with little or no prior experience of delivering psychosocial interventions to mothers with perinatal distress and their new-born infants and have achieved competency to deliver 'Nurturing Care'

intervention following training and practice sessions. These non-specialist facilitators were selected from the universities in the study subdistrict through a job advertisement and interviews.

A cascade model of training and supervision has been followed to deliver 'Nurturing Care' intervention. The training consisted of classroom teaching of basic theoretical concepts of 'Nurturing Care' intervention, its strategies, group discussions and role-plays. The programme trainers trained 15 non-specialist facilitators. The training consisted of 10-days class-room training followed by practice cases under the supervision of programme trainer. The participants of this practice group were pregnant women without distress and their infants, selected from the same community after seeking informed consent for participation in the training. The programme trainer performed live competency rating during practice cases using an adapted version of ENhancing Assessment of Common Therapeutic (ENACT) factors tool for 'Nurturing Care' intervention (Kohrt *et al* 2015). Only competent non-specialist facilitators (mean score above 2.5 on all domains of adapted ENACT) have been allowed

**Table 1** Description of intervention sessions

| Sessions | Content | Timing | Recipients |
|---|---|---|---|
| 1: Introduction and rapport building | ▶ Introducing programme and engaging mother and her family<br>▶ Developing empathetic relationship with mother<br>▶ Ensuring family support system | Beginning of 7th month of pregnancy | Mother and family members |
| 2: Maternal well-being | ▶ Assessment of psychosocial problems of mother<br>▶ Use of guided discovery and behavioural activation for improving mother's health (strategies for improving overall well-being, including diet, rest, relaxation and sleep) | 7th month of pregnancy | Mother and family members |
| 3: Improving mother relationship with significant others | ▶ Thought challenging and behaviour activation using guided discovery (strategies for improving interpersonal relationships and mobilising social support)<br>▶ Awareness about complication during pregnancy | 8th month of pregnancy | Mother and family members |
| 4: Improving mother relationship with infant during pregnancy | ▶ Thought challenging and behaviour activation using guided discovery (strategies for improving bond with the infant during pregnancy. Knowledge and planning of safe delivery and importance of colostrum) | 9th month of pregnancy | Mother |
| 4: Follow-up session after delivery | ▶ Following up after child birth<br>▶ Psychoeducation about child development | Immediate postdelivery | Mother |
| 5: Improving child's physical health | ▶ Thought challenging and behaviour activation using guided discovery (strategies for awareness on benefits of exclusive breastfeeding)<br>▶ Responsive breast feeding<br>▶ Addressing the concept of child crying because of insufficient milk<br>▶ Increasing breast milk by diet and improved frequency-addressing insufficiency of milk<br>▶ Family support for breast feeding<br>▶ Assessment of postpartum psychosocial problems of mother | 8 weeks postnatal | Mother |
| 5A: Improving maternal health | ▶ Assessment of psychosocial problems of mother<br>▶ Use of guided discovery and behavioural activation for improving mother's health | 9 weeks postnatal | Mother |
| 5B: Improving maternal health | ▶ Use of guided discovery and behavioural activation for improving mother's relationship with others | 10 weeks postnatal | Mother |
| 5C: Improving maternal health | ▶ Use of guided discovery and behavioural activation for improving mother's relationship with child | 11 weeks postnatal | Mother |
| 6: Importance of mother–child interaction | ▶ Exclusive breast feeding (EBF) for 6 months<br>▶ Birth spacing<br>▶ Suitable environment-birth spacing<br>▶ Interaction, love and care for child<br>▶ Selecting play activity and conducting with mother and child | 12 weeks postnatal | Mother and infant |
| 7: Follow-up on early stimulation activities | ▶ Revision of activities discussed in session 6<br>▶ Importance of vaccination | 16 weeks postnatal | Mother and Child |
| 8: Awareness on birth spacing; improving child's health | Thought challenging and behaviour activation using guided discovery (strategies for awareness on birth spacing and its relation to child's health)<br>▶ EBF for 6 months<br>▶ Birth spacing<br>▶ Suitable environment-birth spacing<br>▶ Modern methods of contraception<br>▶ Interaction, love and care for child<br>Selecting play activity and conducting with mother and child | 20 weeks postnatal | Mother |
| 9: Introduction to solid food | ▶ Interaction and play with child<br>▶ Selecting play activity and conducting with mother and child<br>▶ Introduction to solid food and concept of super food<br>▶ Food and hygiene<br>▶ Continuation of breastfeeding with weaning at 6 months<br>▶ Importance of vaccination | 24 weeks postnatal | Mother |

Continued

| Table 1 Continued | | | |
|---|---|---|---|
| **Sessions** | **Content** | **Timing** | **Recipients** |
| 10: Introduction to mother–child interaction. Improving mother–child interaction through responsiveness and praise | ► Skill-based activities to promote mother–child interaction<br>► Homework charts are given to monitor the activity<br>► Learning through play activities are reinforced | 28 weeks postnatal | Mother and infant |
| 11: Importance of following child's interest | ► Skill-based activities to promote mother–child interaction<br>► Homework charts are given to monitor the activity<br>► Learning through play activities are reinforced | 32 weeks postnatal | Mother and infant |
| 12: Importance of following child's lead | ► Skill-based activities to promote mother–child interaction<br>► Homework charts are given to monitor the activity<br>► Learning through play activities are reinforced | 36 weeks postnatal | Mother and infant |
| 13: Promote child language development | ► Skill-based activities to promote mother–child interaction<br>► Homework charts are given to monitor the activity<br>► Learning through play activities are reinforced | 40 weeks postnatal | Mother and infant |
| 14: Scaffolding | ► Skill-based activities to promote mother–child interaction<br>► Homework charts are given to monitor the activity<br>► Learning through play activities are reinforced | 44 weeks postnatal | Mother and infant |
| 15: Review of all five strategies | Revision of all strategies and next line of action. Learning through play activities are reinforced. | 48 weeks postnatal | Mother and infant |

to deliver 'Nurturing Care' intervention. Each non-specialist facilitator has been assigned a case load of 8–10 mothers to deliver intervention. The fidelity of the intervention is being evaluated using an adapted ENACT tool developed for the programme. 20% of randomly selected intervention sessions from each non-specialist facilitators at different stages of the intervention delivery will be rated by the programme trainers to assess the fidelity of intervention delivery by non-specialist facilitators.

### Supervision

Programme trainers are experts in perinatal mental health and infant development and provide weekly supervision to non-specialist facilitators to ensure intervention fidelity. Supervision involves discussion of difficulties encountered in delivering intervention delivery and, as well as self-care of the non-specialist facilitators.

### Treatment as-usual (TAU)

As a part of TAU, participants in both study arms receive routine visits by the LHWs of their respective areas. The LHWs are trained to provide preventive maternal and child care including psycho-education about family planning and maternal and child health; provide free of cost contraceptives and antenatal-iron–folic acid supplements throughout the pregnancy starting from second trimester via door to door visits to the households of their allocated areas. LHWs visit new mothers on a monthly basis to monitor the progress of a newborn baby on growth parameters such as weight and height. Moreover, they are given additional duties (provide information about infectious disease or to administer oral polio drops to children) which requires regular visits to houses. In coordination with traditional birth attendants and affiliated nearest health facility, LHWs also provide referral service to mothers for safe motherhood. In the current study,

the number of visits mothers receive by LHWs will be the same for each trial participant regardless of their allocation status.

If during the study's assessments time-points participants in both arms show severe psychiatric disorders that require immediate specialist treatment and follow-up, they will be referred to tertiary mental healthcare facility, Institute of Psychiatry (IoP), Benazir Bhutto Hospital, Rawalpindi. A complete record of services received by trial participants in both arms will be made using the Client Services Receipt Inventory (CSRI).

### Outcome measures

The effectiveness of intervention will be evaluated on a range of caregiver and child outcomes. The outcome measures will be administered by the trained assessment team blind to the allocation status of participants at baseline (at the time of recruitment, ie, third trimester of pregnancy), 6 and 12 months' postpartum. The self-report outcome measures will be orally administered by the trained assessment team blind to the allocation status of the participants. The detail of the outcome measures is given below (see table 2 for schedule of assessments).

#### Primary outcome

Caregivers–child interaction: maternal responsive caregiving behaviours will be assessed at 12 months' postpartum across two-arms (primary endpoint) using the OMCI tool.[22] OMCI tool is based on responsive parenting framework proposed by Landry and colleagues.[23] The tool comprises of 19 items related to maternal and child behaviours. Maternal behaviours include maternal affect, maternal touch, maternal verbalisation, sensitivity and contingent responses, scaffolding, language stimulation and focus; whereas, child outcomes include, child affect, child focus, child's communication efforts and

**Table 2** Schedule of assessments

| S. no. | Outcomes | Screening | Baseline third trimester of pregnancy | 6 months postpartum | 12 months postpartum |
|---|---|---|---|---|---|
| | | | **Time points** | | |
| *Primary outcome* | | | | | |
| 1. | OMCI | | | | X |
| *Secondary outcomes* | | | | | |
| 2. | SRQ | X | X | X | X |
| 3. | WHO-QoL | | X | | X |
| 4. | MSPSS | | X | | X |
| 5. | WHODAS | | X | | X |
| 6. | Maternal empowerment-brief questionnaire | | X | | X |
| 7. | BSID-III | | | | X |
| 8. | ASQ3 | | | | X |
| 9. | Home inventory | | | | X |
| 10. | Weight and height measurements | | | | X |
| 11. | Early breastfeeding-brief questionnaire | | | X | |
| 12. | Diarrhoeal episodes-brief questionnaire | | | X | X |
| 13. | Immunisation-record form | | | X | X |
| 14. | CSRI | | | | X |

ASQ3, Ages and Stages Questionnaire-3; BSID-III, Bayley Scale of Infant Development-III; CSRI, Client Service Receipt Inventory; MSPSS, Multidimensional Scale of Perceived Social Support; OMCI, Observation of Mother–Child Interaction; SRQ, Self-reporting Questionnaire; WHODAS, WHO Disability Assessment Schedule; WHO-QoL, WHO, Quality of Life.

mutual enjoyment. The scoring is done by recording the frequency of the occurrence of each of the above-mentioned behaviours during the mother–child interaction. The sum is calculated by adding the scores of mother and child domains; where higher score indicates more responsive interactions. The tool has been designed to live rate mother–child interaction of 2 years old toddlers in low resource settings of rural Pakistan. It has been extensively used in similar population in Pakistan and found to be sensitive to detect change in mother–child interaction as a result of receiving responsive caregivers intervention at 12 and 24 months postpartum.[22] In the present study, the research assistants will observe a live 5 min mother–infant interaction while mother and infant will play together with a picture book.

### Infant secondary outcomes

1. *Bayley's Scales of Infant Development (BSID III)*[24] will be used to measure cognitive, language, motor, socio-emotional development and adaptive behaviour at 12 months' postpartum. The tool has been used previously in similar South Asian settings.[25 26]
2. *Ages and Stages Questionnaire 3rd Edition (ASQ3)*[27] will be used to measure developmental delays in communication, gross motor, fine motor, problem solving and personal social domains at 12 months' postpartum. It provides cut-off score in five domains of development that indicates possible need for further evaluation. It

highlights results that fall in a 'monitoring zone', to make it easier to keep track of children at-risk. The tool has been validated to use in South Asian community settings.[26 28]

3. Home environment measured by *Home inventory*:[29] quality and quantity of stimulation and support in the child's home will be measured using home inventory at 12 months' postpartum. The tool has been validated to use in Pakistan.[30]
4. *Anthropometric measurements*:[31] infants in both groups will be weighed and measured with standard techniques at 12 months' postpartum. Growth data will be converted into SDs (Z scores) for weight and length with WHO Anthro (V.3.2.2).
5. *Early breastfeeding (EBF)*:[32] assessors will document what the infant was fed in the last 24 hours at 6 months postpartum. This information will be categorised as either EBF, partial breastfeeding (giving an infant some breastfeeding, and some additional foods, either milk, cereal or other food items) or no breastfeeding. If the infant will not exclusively breastfeed, details of when (age of infant) and why EBF was discontinued will be recorded. At the first 6 months follow-up, details about early initiation, discarding colostrum, use of prelacteals and reasons for delaying or not initiating breastfeeding will be assessed.

6. The number of *diarrhoeal episodes* in the infants during the 2 weeks before interview will be recorded at 6 and 12 months' postpartum, with a questionnaire used in previous studies.[33] Diarrhoea will be defined as three or more unformed stools passed in 24 hours, and a diarrhoeal episode will be defined as being separated from another episode by at least three diarrhoea-free days.

7. Records of *immunisation* will be assessed for all infants in the study, and infants will be classified as those with or without up-to-date immunisation status 6 and 12 months' postpartum.

### Maternal secondary outcomes

8. Maternal distress will be measured by the *SRQ*[34] at baseline, 6 and 12 months postpartum. SRQ is a 20-item self-report measure to detect non-specific psychological distress, developed by the WHO. Psychological distress is represented by subscales of depression/anxiety, somatic symptoms, reduced vital energy and depressive thoughts. The SRQ items are scored 0 or 1. A score of 1 indicates the presence of symptoms of psychological distress during past month and a score of 0 indicates the absence of symptoms. The maximum score indicates the presence of higher psychological distress. A score of >9 indicates increased risk of maternal distress/postpartum depression.[8 35]

9. Caregivers' health-related quality of life will be measured by *WHO Quality of Life (WHOQoL)-Brief version* at baseline and 12 months postpartum. WHOQoL assess the participants position in life in the context of the culture and value systems in which they live and in relation to their goals, expectations, standards and concerns. WHOQoL comprises of four domains namely physical health, psychological health, social relationships and environment. Items are rated on a 5-point Likert scale, where 1 represents 'disagree' or 'not at all' and 5 represents 'completely agree' or 'extremely'.

10. The level of social support will be measured using *Multidimensional Scale of Perceived Social Support (MSPSS)*[36] will be measured baseline and 12 months' postpartum. It includes 12 items which cover three dimensions: support from family, friends and significant other. Each item is rated on a 7-point Likert-scale (1=*very strongly disagree*; 7=*very strongly agree*). A total score is calculated by summing the responses of all items (range 12–84) with higher scores indicating higher levels of perceived social support.

11. Caregiver's level of functioning will be measured using *WHO Disability Assessment Schedule (WHODAS)*[37] at baseline and 12 months' postpartum. WHODAS assesses participants' health-related difficulties in the level of functioning in six domains of life (understanding and communicating; moving and getting around; attending to one's hygiene, dressing, eating and staying alone; interacting with other people; domestic responsibilities, leisure, work and school and joining in community activities, participating in society) over the past 30 days.

12. *Maternal financial empowerment*: women's empowerment will be measured at baseline and 12 months postpartum, using a questionnaire consisting of: (a) two items previously used in the same population[38] that assess financial empowerment (ie, whether a woman is given a lump-sum of money for day-to-day expenses and whether she can make independent decisions about its use). Women who answer 'yes' to both items on financial empowerment will be classified as 'empowered'.

13. Utilisation of healthcare will be measured by CSRI adapted for perinatal population of Pakistan, at 12 months postpartum.[32]

As the study is being conducted amidst COVID-19, all in-person assessments will be conducted by following appropriate precautionary measures.

### Randomisation and blinding

The participants were randomised into intervention or control arm on a 1:1 basis using computerised software by an independent researcher. The allocation concealment was ensured by keeping random sequence in sequentially numbered, opaque, sealed envelopes, at the off-site centre. Only assessment team (conducting baseline and follow-up assessment) will be blind to the treatment allocation of trial participants. To maintain the blinding, all trial participants will be briefed, prior to the assessment about the significance of blinding and the importance of not disclosing their allocation status to the assessment team members. If unblinding does occur, this will be documented, the assessment will be halted and a new assessor will be assigned to complete the assessment.

### Safety measures

In the present study, a referral pathway, based on the existing referral system, has been established to ensure the safety of the study participants. The IoP is the tertiary mental healthcare facility in the north of Pakistan and situated within the public tertiary healthcare facility and teaching hospital, the Benazir Bhutto Hospital, Rawalpindi. It caters to the mental health needs of the populations in the metropolitan city of Rawalpindi and is the specialist referral facility for common mental health problems identified at the adjacent primary healthcare centres. In our study, if a participant discloses any event of abuse, intimate partner violence or is at imminent risk of harm to herself and others, identified at the time of assessment or during the intervention delivery, the research team will record the event and referral to the IoP will be made for appropriate medical, medico-legal and psychosocial assessment and rehabilitation by a multidisciplinary team led by qualified psychiatrists.

## Adverse events reporting

All adverse events (AEs) and serious AEs reported by the trial participants, their husband and in-laws of recruited participants, LHWs or observed by the investigator or the research team (assessment and intervention team) will be recorded by data manager. AE is defined as any undesirable experience occurring to a subject during the time frame of the study, whether or not considered relevant to the research procedure. All AEs will be collected by assessment team at 12 months' postpartum follows and will be reviewed periodically by trial steering committee (TSC) which will determine the necessary steps to be taken with respect to the ongoing trial.

## Data management

The baseline and follow-up assessments data will be collected by the assessment team on android-based application using hand held devices (tablets). Data will be stored at the secure web server and will be checked for consistency and quality at the end of each day. All data sets will be stored in password-protected computer, only accessible by the responsible individual. The data collected in hard copies (collected by the interviewers) will be safely stored in locked cabinets. Qualitative data will be fully anonymised and coded. No identifying names or details will be recorded.

## Statistical analysis

The findings will be reported according to the Consolidated Standards of Reporting Trials guidelines for randomised controlled trial[39] (figure 1). This will include the flow of research participants through each stage of the trial, including the number eligible, randomly assigned, receiving the intended treatment, completing the study protocol and analysed for the primary outcome. The primary analyses will be on intent-to-treat basis consisting of all participants included according to the arms in which they are randomised and secondary analyses will be based on per-protocol population.

For the analysis of the primary outcome (caregiver–child interaction across two-arms at 12 months postpartum), a linear regression model will be employed with treatment as the sole predictor. In addition, adjusted linear model analysis will be performed with the prespecified covariates (gender of infant, mother's education and severity of mothers' psychosocial distress measured using SRQ) measured at baseline being added into the above linear model. The crude and adjusted mean differences in the primary outcome together with its 95% CIs at 12 months will be derived from the linear models. In addition, subgroup analysis of primary endpoint will be performed on the above prespecified covariates.

Analysis of secondary continuous outcomes with single follow-up measurement will be done in a similar fashion as the primary endpoint analysis. Analysis of secondary continuous outcomes with repeated follow-up measurement will be performed using a linear mixed model with treatment visit, interaction between treatment and visit as fixed effects, the baseline value of the outcome as covariate, if it is available and subject as random effects. The analysis of binary outcomes will also use a generalised linear/mixed model depending on whether there will be a repeated measurement. ORs with their 95% CIs will be derived from the generalised linear/mixed model analysis. Missing primary outcome and secondary outcomes with a single measurement will not be imputed; whereas, missing secondary outcomes with repeated measurements will be imputed in sensitivity analysis using the last observation carried forward strategy. Missing baseline covariates will be imputed using simple imputation methods in the covariate adjusted analysis based on the covariate distributions in the sample. For a continuous variable, missing values will be imputed from random values from a normal distribution with mean and SD calculated from the available sample. For a categorical variable, missing values will be imputed from random values from a uniform distribution with probabilities $P_1$, $P_2$, … and $P_k$ from the sample. Seed for the imputation will be set as the date of data analysis. The study is powered to make a single comparison in the primary outcome at 12 months only. All analyses will be described in detail in the statistical analysis plan. SAS V.9.4 and SPSS V.21 will be employed for the statistical analyses.

## Potential mediators

We will explore pathways through which the intervention may influence primary outcome (ie, mother–child interaction). There is strong empirical evidence that the quality of the mother–child interactions, such as her sensitivity and responsiveness, is associated with both maternal depression and child development.[40–43] There is also some evidence that perinatal depression interventions, especially ones that include a parenting component, can improve maternal distress; maternal responsiveness, resulting in improved developmental outcomes.[44] We will assess maternal responsiveness with the maternal items of OMCI that measure maternal responsive behaviours during mother–child interaction. The OMCI has been used to evaluate the impact of maternal interventions on maternal responsiveness.[45] Maternal distress will be assessed using SRQ. The tool has been used extensively in the study settings previously to measure maternal distress.[46]

*Social support*: low social support is a consistent predictor of mental ill health among mothers.[47 48] Social support can be theorised to be either a mediator or effect modifier; since building social support is a key feature of the integrated intervention, we hypothesise that increased social support will mediate the relationship between the intervention and child outcomes. *Maternal empowerment*: maternal empowerment will be assessed using two items previously used in the same population.[38] *Breastfeeding and other health-related variables*: perinatal depression interventions similar to this integrated intervention have

resulted in higher rates of exclusive breastfeeding at 6 months and increased likelihood of completion of scheduled immunisations.[6] These health-related behaviours may contribute to the positive effect of intervention on improving maternal well-being and infant development, we will assess the impact of behaviours related to feeding (breastfeeding, introduction of solid food) and immunisations on mother–child interaction.

## Cost–effectiveness analysis

Health economic analysis will be conducted to determine the difference in costs and outcomes in the intervention as compared with the control arm. The cost of the intervention will be calculated based on the sum of materials, personnel training and delivery. We will then compare the combination of the cost of the intervention and the service cost data with mother–child interaction summary scores. If the intervention results in more favourable summary scores (improvement in mother–infant interaction) and lower total costs, it will be described as superior to no intervention option. If, in contrast, the intervention results in more favourable scores but is also more expensive, then the cost–effectiveness calculation will be based on the cost and scores of mother–infant interaction outcomes.

Primary analysis will be of total costs over the 12 months postpartum follow-up treatment period. Recognising that cost data are often skewed, the bootstrap technique will be applied. The sampling with replacement from original observed pairs of costs and effects will be employed to maintain correlation structure between costs and benefits, and bootstrapping sampling will be repeated 1000 times. For each bootstrap sample, an estimate of differential total mean costs and expected mean effectiveness will be calculated. The 95% CIs for the differential estimates will be derived from the 25th and 97.5th percentiles.[49 50]

Cost–effectiveness will be assessed by combing costs with the primary outcome measure (ie, mother–infant interaction) scores in incremental cost–effectiveness analysis. Repeat resampling from the costs and effectiveness data (bootstrapping) will be used to calculate the probability that each of the treatment is the optimal choice, subject to a range of possible maximum values (ceiling ratio) that a decision-maker might be willing to pay for a unit improvement in mother–infant interaction scores. The results of the cost–effectiveness will be reported as incremental cost–effectiveness ratios, and acceptability curves which summarise the information contained in a cost–effectiveness plane.[51]

## Mixed-methods evaluation

We will evaluate the feasibility of our intervention programme using mixed-methods study design. Mixed-methods evaluation of implementation will be carried out to assess assumptions underlying the intervention strategy and will cover other aspects of programme implementation such as number of research participants who signed-up and complete the intervention and satisfaction with intervention, difficulties and successes in carrying out intervention activities, acceptability, feasibility and appropriateness of using integrated intervention.

Satisfaction with the intervention and barriers and facilitators to implementation, feasibility of workload and difficulties and successes in carrying out intervention activities and satisfaction with training and supervision will be explored through semi-structured interviews with the relevant stakeholders.

## Ethics

Ethics approval for the present study was obtained from the Human Development Research Foundation Institutional Review Board, Islamabad Pakistan. Informed consent will be obtained from all participants prior to data collection. Privacy and confidentiality of the research participants will be ensured throughout the study. Trial management will be provided by TSC that comprises principal investigator (PI), coinvestigator (s), trial coordinator, senior researchers and intervention staff who will meet monthly.

## Dissemination

There is a growing evidence base on the feasibility, acceptability and effectiveness of peer/parent delivered, psychosocial and ECD interventions.[6 18 52] Training and supervision of large numbers of health workers in new interventions is a key barrier to scale up of programmes. Our intervention combines evidence-based elements into a single package, making delivery more effective and synergistic, through a single delivery agent that is, non-specialist facilitator. The use of non-specialist facilitators in disseminating the interventions has the potential to extend the reach of the intervention in a cost-effective fashion. If proven effective, the study will contribute to scale-up care for maternal and child mental health in low resource settings, globally. The PI, trial coordinator and trial statistician will have access to the clean data sets. Deidentified trial data can be accessed by contacting PI on reasonable request. We will disseminate our results to key stakeholders. Dissemination activities will include publication of study results in open access, peer-reviewed journals and presentations at academic conferences and community forums.

**Author affiliations**
[1]Implementation Science, Human Development Research Foundation, Islamabad, Punjab, Pakistan
[2]Department of Primary Care and Mental Health, University of Liverpool Faculty of Health and Life Sciences, Liverpool, UK
[3]Global Institute of Human Development, Shifa Tameer-e-Millat University, Islamabad, Pakistan
[4]Department of Clinical Sciences, Liverpool School of Tropical Medicine, Liverpool, UK

**Contributors** SUH, ZeH and DW will have the access of all data sets. ZeH drafted the initial manuscript. SUH conceived the study and wrote the final manuscript. AG,

FS, AR, BM, HJ, DW and AR contributed to the writing. All authors approved of the final version of the manuscript.

**Funding** This work is supported by Grand Challenges Canada, Saving Brains, Grant Number SB-POC-1809-18680.

**Competing interests** None declared.

**Patient and public involvement** Patients and/or the public were involved in the design, or conduct, or reporting or dissemination plans of this research. Refer to the Methods section for further details.

**Patient consent for publication** Not required.

**Provenance and peer review** Not commissioned; externally peer reviewed.

**ORCID iDs**
Zill-e- Huma http://orcid.org/0000-0002-2681-1771
Duolao Wang http://orcid.org/0000-0003-2788-2464

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
