## [Reviewer comments · BMJ Open]

ARTICLE DETAILS

TITLE (PROVISIONAL)	Evaluating the impact of a common elements based intervention to improve maternal psychological wellbeing and mother-infant interaction in rural Pakistan: Study protocol for a randomized controlled trial
AUTHORS	Huma, Zill-e-; Gillani, Ayella; Shafique, Fakhira; Rashid, Alina; Mahjabeen, Bushra; Javed, Hashim; Wang, Duolao; Rahman, Atif; Hamdani, Syed Usman

VERSION 1 – REVIEW

REVIEWER	Pengcheng Xun Indiana University Bloomington
REVIEW RETURNED	07-Feb-2021

GENERAL COMMENTS	Major concerns: 1. The current null hypothesis is effect size ≥ 0.4, and the alternative hypothesis is effect size < 0.4. Then it is a one-sided test. The authors should realize this and give strong justifications for using a one-sided test than a two-sided. Usually, the effect size is not needed to be emphasized in the hypothesis.2. Without the background information (e.g., the mean and standard deviation of the score), this effect size of 0.4 is less meaningful or not easily understood.3. The authors should also clarify which test was used in the power justification sentence? "A two-sided hypothesis test" is not enough. Based on this verification, a t-test should have been used.4. The authors should further clarify the primary outcome. Should give one or two sentences to describe what the score looks like. E.g., how many domains/dimensions does it include? What is the theoretical range of the score from 19 items? What is the normal range of this score in practice? Some critical parameters for validation should also be mentioned instead of just saying sth like "the tool has been validated."5. It is not clear how the recruitment process will guarantee the sample's representativeness.6. The authors mentioned, "The data will be collected at baseline, 6- and 12-months' post-partum." It is not very clear the meaning of "baseline" here; does it mean at the recruitment or right after giving birth to a baby?7. It is not clear what the baseline data and 6-month post-partum data will be used together with the 12-months/ post-partum data?
---

	8. As a proposal, the authors should be very clear about what analyses will be conducted? Which models or methods will be used? Will the “multiple comparison/testing” be a concern since there are many outcomes? If yes, how to tackle it? How to take care of the potential missing values in the data? Minor concerns:  1. There are some language problems in this manuscript, e.g.:  a. Evidence based should be evidence-based. b. “A separate and overlapping packages” should be “A separate and overlapping package.” c. “A two arm, single blind...” should be “A two-arm single-blinded...”. d. “randomized” and “randomised” are mixed in the text. e. Etc. 2. Page 2, line 53 to 54: in the sentence “...the intervention is being delivered using blended methodology (telephonic delivery as well as individual sessions where possible.”:  a. “blended methodology” should be “a blended methodology.” b. “methodology” is a too big word in this context. c. Needs to make sure “telephonic delivery” and “individual sessions” are exclusive to each other. Not very clear.
--	--

REVIEWER	Katie Rose Sanfilippo Goldsmiths University of London, Psychology
REVIEW RETURNED	25-Feb-2021

GENERAL COMMENTS	Overall, this protocol lays out very clearly the plans of a well-considered and potentially highly impactful RCT. I appreciate all the work the researchers have put into this protocol submission. I found the common elements based approach to be very interesting and believe that others in this field will agree. I have only minor comments and believe that this protocol should be accepted as it will be of interest to many BMJ Open readers. My minor corrections/suggestions are laid out below:  1. Make sure to include the dates of the study within the manuscript, potentially within the Trial Registration section of the manuscript 2. The last point of the article summary: “Exclusion of women with perinatal depression may limit the generalizability of findings” is unclear and inconsistent with the information presented in the rest of the protocol. From what I understood women who scored a 9 or above on the SRQ-20 were included. While not a diagnosis, women with depression could be included in that group. More clarity is needed surrounding what this point is trying to say. 3. It was unclear until the description of the intervention table that the intervention started in pregnancy. It would be helpful to bring this point out earlier and even discuss some of the potential benefits of starting during the antenatal period. One way to do this would be to include some brief information in the introduction about the potential risks of antenatal stress on foetal development and infant cognitive and developmental outcomes (could include work by Prof Vivette Glover for example). I believe starting in pregnancy is a real benefit of the intervention and therefore should be highlighted. 4. The intervention includes 15 sessions across one year which is extensive. It might help to include a short justification for why this intervention duration was chosen. You mention briefly potential issues this might create in terms of scale-up and cost. Therefore, I think more explanation on why this long intervention period was
--

	chosen would be beneficial. 5. You mention that participants in the TAU group who show signs of severe psychiatric disorders will be referred on to a tertiary mental health facility. What about participants in the intervention group? Will they also be referred on? Generally, the protocol would benefit from more information about other ethical considerations made, especially around disclosure. I know item 17 of the SRQ-20 asks if participants have thought of ending their life. What are the processes in place if women answer yes to this item or disclose that they have thought of or attempted suicide? Also, what if a woman discloses intimate partner violence within the sessions? Some explanation of how these will be managed would be helpful to include. 6. How will the self-report measures be administered? Will they be administered on paper, through tablets or orally by trained research assistants? This just needs to be made clear. 7. When you discuss potential mediators you mention maternal distress but do not give a hypothesis or rationale as to why you would investigate this as you do for the other potential mediating variables. It would help with consistency and clarity to include this information. 8. For Figure 1, I would suggest including when in pregnancy women will be enrolled and when the baseline measurement is taken (e.g. 3rd trimester). Overall, I really enjoyed reading your work. Thank you again for putting together this wonderful and important research and I look forward to reading the final results of the trial.
--	--

REVIEWER	Lara Pierce Boston Children's Hospital
REVIEW RETURNED	26-Feb-2021

GENERAL COMMENTS	The manuscript under review presents a protocol for a randomized controlled trial of an intervention aimed at improving maternal psychological wellbeing and mother-infant interaction in rural Pakistan. Improvements in these variables are expected to improve developmental outcomes for children in the long term. The intervention employs a common elements approach, which can be flexibly applied in low-resource settings, and which can be provided by trained non-specialist providers (e.g., community health workers). The intervention was developed in collaboration with stakeholders in the local communities, which is a key strength. By partnering with families and the community, and capitalizing on an existing system (e.g., regular visits to new mothers by community health workers), this intervention is likely to be successfully implemented. I provide some comments and questions below: Title  - "Evaluating the impact of a common elements based intervention to improve the maternal..." "the" should be removed from the title. Article Summary  - Mixed methods process evaluation is described to assess blended methodology required as a result of COVID-19, however I do not see further discussion of this in the manuscript. It would be helpful for this, as well as other COVID-19 considerations, to be discussed in another section of the article aside from just the summary. - It is noted that exclusion of women with perinatal depression may limit the generalizability of findings, however it is not clear to me
--

when/how women with perinatal depression are excluded. Moreover, the manuscript seems to be framed around addressing perinatal depression as an intervention target, which suggests to me that these women should be included. Could the authors clarify this point?

Introduction

- Could the authors please define the common elements approach, as it is a term not all readers will be familiar with. More detail surrounding from where packages of care were chosen, and why, would also be beneficial.
- The phrasing “the mother, in turn, requires...via a well monitored and supervised delivery system” suggests to me that mothers need to have very specific training to effectively care for children, which I do not think the authors mean to imply. Would replacing “required” with “receives” retain the meaning the authors intended?

Method

- Could the authors provide the dates that the intervention did/will take place?
- How was the estimate of 20% attrition determined?
- Given that the SRQ reflects psychological distress across domains (e.g., including depression, anxiety), can the authors clarify whether a score >9 indicates risk of postnatal depression specifically, or whether it represents a more general mental health risk.
- Are any additional resources provided to mothers who score >9 on the SRQ, regardless of treatment arm? I am wondering whether there is a clinically significant cut-off beyond which it is ethical practice to provide some type of referral or intervention.
- Can the authors describe how these particular common elements (e.g., WHO, UNICEF, etc.) were chosen for this intervention, as opposed to others?
- Can the authors provide examples for “guided discovery using pictures” and “behavioural activation”?
- I am curious about the decision to introduce the breastfeeding module at 8 weeks postpartum. I could see benefit to introducing this module at the follow-up after delivery given that that is when the breastfeeding relationship is likely to be established. Can the authors clarify the rationale for including this only at a later point?
- Can the authors include information about how many LHW visits mothers receive, and the timeline for those visits? Would they typically extend through the first postpartum year (as the intervention does) or are there differences in visit schedule across intervention and TAU groups?
- Can the authors clarify whether/how baseline differences in mother-child interactions between intervention and TAU groups will be assessed, given that the OMCI is only administered at the end of the intervention?
- Could the authors discuss any validation/norming of the BSID III and ASQ3 that is relevant for cross-cultural administration?
- Under “Early breastfeeding (EBF)” I believe the term “artificial foods” is incorrect as it refers to “milk, cereal, or other food items” which are not artificial.
- The authors state that “missing data will be treated as missing at random,” but can that be determined prior to data collection? How will the authors determine that data are missing at random, and will comparisons between those with and without data be compared on other measures to determine systematic differences between those who did and did not complete a measure?

	- Potential mediators: Can the authors clarify why maternal responsiveness is assessed using the HOME inventory as opposed to (or in addition to) being coded from the mother-child interactions? Given that potential mediators and child outcomes are measured at the same time point, can the direction of effects be inferred? For example, if more responsive interactions and higher cognitive scores are observed, could it alternatively be the case that infants with higher cognitive scores elicit more responsive interactions? SPiRiT Checklist - “Strategies for achieving adequate participant enrolment to reach target sample size” and “Plans to promote participant retention and complete follow-up, including list of any outcome data to be collected for participants who discontinue or deviate from intervention protocols” are both marked as N/A. However, participant enrolment and retention through a longitudinal study of this nature is critical to its success. Could the authors describe strategies to achieve adequate enrolment and retention? - “Criteria for discontinuing or modifying allocated interventions for a given trial participant (eg, drug dose change in response to harms, participant request, or improving/worsening disease)” is marked N/A. Are there circumstances under which the intervention would be discontinued? (e.g., conditions developed by mother or baby, lack of adherence to protocol, etc.)?
--	---

VERSION 1 – AUTHOR RESPONSE

Reviewer: 1

Dr. Pengcheng Xun, Indiana University Bloomington
Comments to the Author:

Major concerns:

Comment 1: The current null hypothesis is effect size ≥ 0.4 , and the alternative hypothesis is effect size < 0.4 . Then it is a one-sided test. The authors should realize this and give strong justifications for using a one-sided test than a two-sided. Usually, the effect size is not needed to be emphasized in the hypothesis.

Response: We are thankful to the reviewer for his comment!

In the current study, we are using two-sided hypothesis test as mentioned in the “sample size calculation” section (page 5, line 193-194). We did not mention it in the hypothesis that is omission at our part and we are regretful for the inconvenience caused.

As suggested by the reviewer, we have removed the effect size from the hypothesis (page 4, line 116-117) and now only kept it in the sample size calculations section.

Comment 2: Without the background information (e.g., the mean and standard deviation of the score), this effect size of 0.4 is less meaningful or not easily understood.

Response: We are thankful to the reviewer for his comment!

The effect size for the present study was estimated based on the findings of a similar study conducted in Pakistan, where OMCI was used as one of the outcomes to evaluate the effect of responsive stimulation intervention on caregiver-child interaction at 12 and 24-months post-partum (Yousafzai et al, 2015). The findings of the study showed mother-child dyads receiving responsive caregiving intervention had significantly higher mean scores on OMCI at 12 months (mean 32.3 [SD 8.3] vs mean 27.1 [SD 8.2], $p < 0.0001$) than those who were allocated to control arm. Moreover, repeated-

measures analysis showed that effects were sustained at 24-months. As the effect sizes for OMCI in Yousafzai et al 2015's study ranged between 0.6 to 0.9; therefore, a conservative effect size of 0.4 (two-sides hypothesis) at 12-months post-partum was proposed in the present study. The same above mentioned text has been added in the manuscript (please see page 5, line 180-190 of the manuscript).

Reference:

- Yousafzai, A. K., Rasheed, M. A., Rizvi, A., Armstrong, R., & Bhutta, Z. A. (2015). Parenting skills and emotional availability: an RCT. *Pediatrics*, 135(5), e1247-e1257.

Comment 3: The authors should also clarify which test was used in the power justification sentence? "A two-sided hypothesis test" is not enough. Based on this verification, a t-test should have been used.

Response: We are thankful to the reviewer for his comment!

In the present study, we have used the two-sided t-test for the sample size calculation.

The required information has been added in the manuscript too (please see page 5, line 192 of the manuscript).

Comment 4: The authors should further clarify the primary outcome. Should give one or two sentences to describe what the score looks like. E.g., how many domains/dimensions does it include? What is the theoretical range of the score from 19 items? What is the normal range of this score in practice? Some critical parameters for validation should also be mentioned instead of just saying sth like "the tool has been validated."

Response: We are thankful to the reviewer for his comment!

The required information has been added on page 11, line 300-315 of the manuscript reproduced here for ease.

Maternal responsive caregiving behaviours will be assessed at 12-months' postpartum across two arms (primary endpoint) using the Observation of Mother and Child Interaction (OMCI) tool [20]. OMCI tool is based on responsive parenting framework proposed by Landry and colleagues [21]. The tool comprises of 19 items related to maternal and child behaviors. Maternal behaviors include maternal affect, maternal touch, maternal verbalization, sensitivity and contingent responses, scaffolding, language stimulation and focus; whereas, child outcomes include, child affect, child focus, child's communication efforts, and mutual enjoyment. The scoring is done by recording the frequency of the occurrence of each of the above mentioned behaviors during the mother-child interaction. The sum is calculated by adding the scores of mother and child domains; where, higher score indicates more responsive interactions.

The tool has been designed to perform live rating of mother-child interaction in low resource settings of rural Pakistan. It has been extensively used in similar population in Pakistan and found to be sensitive to detect change in mother-child interaction as a result of receiving responsive caregivers intervention at 12 and 24-months post-partum [20]. In the present study, the research assistants will rate a live 5-minute mother-infant interaction while mother and infant will play together with a picture book.

Comment 5: It is not clear how the recruitment process will guarantee the sample's representativeness.

Response: We are thankful to the reviewer for his comment!

We agree with the reviewer's comment and have added the details in the recruitment procedure to mention how the recruitment process will guarantee the sample's representativeness (please see page 5-6, line 197-210 of the manuscript). The relevant text has been reproduced here for ease.

Recruitment procedure:

Before recruiting participants in the trial, the local community stakeholders were taken on-board. As mentioned in the study settings, the current study is being conducted in the rural sub-district of Gujar Khan, Rawalpindi district, Pakistan. The study sub-district consists of 36 Union Councils (UCs)- the smallest administrative units of the sub-district. Each UC has a Basic Health Unit (BHU) that provides primary health care to about 15,000–20,000 local population. Each BHU is staffed by a doctor, midwife, vaccinator, and 15–20 village-based Community Health Workers (CHWs) also called Lady Health Workers (LHWs). These women have completed secondary school and are trained to provide preventive maternal and child health care, and education in the community. Each LHW is responsible for about 100-150 households in her village and keeps a register of every new pregnancy in her catchment area. The research participants in the present study are being recruited from the catchment areas of LHWs, where, the research team has assessed every woman in the third trimester of their pregnancy to evaluate their eligibility for participation in the trial. 250 pregnant women in their third trimester, who fulfil the eligibility criteria, are being enrolled in the trial and individually randomized into intervention and TAU arms (1:1 ratio) (please refer to Fig 1- CONSORT Flow).

Comment 6: The authors mentioned, “The data will be collected at baseline, 6- and 12-months’ post-partum.” It is not very clear the meaning of “baseline” here; does it mean at the recruitment or right after giving birth to a baby?

Response: The baseline in the present study refers to the time of recruitment (i.e. 3rd trimester of pregnancy). We have clarified it in the manuscript as well. Please see page 11, line 292 of the manuscript.

Comment 7: It is not clear what the baseline data and 6-month post-partum data will be used together with the 12-months/ post-partum data?

Response: We are thankful to the reviewer for his comment!

The 6-month post-partum data will be used to do an exploratory mediation analysis to evaluate the impact of maternal distress on primary outcome i.e. mother-infant interaction.

Comment 8: As a proposal, the authors should be very clear about what analyses will be conducted? Which models or methods will be used? Will the “multiple comparison/testing” be a concern since there are many outcomes? If yes, how to tackle it? How to take care of the potential missing values in the data?

Response: We are thankful to the reviewer for his valuable suggestion!

We have revised the analysis plan as indicated (please see page 15-16, line 463-497). The text has been reproduced here for ease.

For the analysis of the primary outcome (caregiver-child interaction across two arms at 12-months post-partum), a linear regression model will be employed with treatment as the sole predictor. . In addition, adjusted linear model analysis will be performed with the pre-specified covariates (gender of infant, mother’s education and severity of mothers’ psychosocial distress measured using SRQ) measured at baseline being added into the above linear model. The crude and adjusted mean differences in the primary outcome together with its 95% confidence intervals at 12-months will be derived from the linear models. In addition, subgroup analysis of primary endpoint will be performed on the above pre-specified covariates.

Analysis of secondary continuous outcomes with single follow-up measurement will be done in a similar fashion as the primary endpoint analysis. Analysis of secondary continuous outcomes with repeated follow-up measurement will be performed using a linear mixed model with treatment visit,

interaction between treatment and visit as fixed effects, the baseline value of the outcome as covariate if it is available, and subject as random effects. The analysis of binary outcomes will also use a generalized linear/mixed model depending on whether there will be a repeated measurement. Odds ratios with their 95% confidence intervals will be derived from the generalized linear/mixed model analysis.

Missing primary outcome and secondary outcomes with a single measurement will not be imputed but missing secondary outcomes with repeated measurements will be imputed in sensitivity analysis using the last observation carried forward strategy. Missing baseline covariates will be imputed using simple imputation methods in the covariate adjusted analysis based on the covariate distributions in the sample. For a continuous variable, missing values will be imputed from random values from a normal distribution with mean and SD calculated from the available sample. For a categorical variable, missing values will be imputed from random values from a uniform distribution with probabilities P1, P2, ..., and Pk from the sample. Seed for the imputation is set as the date of data analysis (eg, 270421).

The study is powered to make a single comparison in the primary outcome at 12 months only and other comparisons are exploratory in nature. Therefore, there is no multiplicity issue. All analyses will be described in detail in the statistical analysis plan. SAS 9.4 and SPSS 21 will be employed for the statistical analyses.

Minor concerns:

Comment 9: There are some language problems in this manuscript, e.g.:

Comment 10: Evidence based should be evidence-based.

Response: Changes have been made to following pages;

Page 2, line 34-35

Page 3, Line 98 & 102

Page 6, line 213

Comment 10: "A separate and overlapping packages" should be "A separate and overlapping package."

Response: The text has been revised as indicated (Please see page 2, line 36)

Comment 11: "A two arm, single blind..." should be "A two-arm single-blinded..."

Response: The text has been revised as indicated:

Please see

Page 2, line 38 & 64

Page 4, line 124

Comment 12: "randomized" and "randomized" are mixed in the text.

Response: The text has been revised as indicated. Please see page 2 (line 64)

Comment 13: Page 2, line 53 to 54: in the sentence "...the intervention is being delivered using blended methodology (telephonic delivery as well as individual sessions where possible.):

a. "blended methodology" should be "a blended methodology."

b. "methodology" is a too big word in this context.

c. Needs to make sure "telephonic delivery" and "individual sessions" are exclusive to each other. Not very clear.

Response: We are thankful to the reviewer for his valuable suggestions!

The text has been revised as indicated. Please see page 2, line 68-69. The text has been reproduced here for ease.

The COVID-19 pandemic has resulted in limited face to face contact with the study participants and the intervention is being delivered via telephone or in-person where possible.

Page 6, line 236-238: During the COVID-19 pandemic, when the in-person sessions are not possible, we are delivering the scheduled intervention session via telephone. All those sessions delivered via telephone are not repeated in-person.

Reviewer: 2

Ms. Katie Rose Sanfilippo, Goldsmiths University of London
My minor corrections/suggestions are laid out below:

Comment 1: Make sure to include the dates of the study within the manuscript, potentially within the Trial Registration section of the manuscript

Response: We are thankful to the reviewer for her comments!
The study dates have been added as suggested (please see page 2, line 57).

Comment 2: The last point of the article summary: "Exclusion of women with perinatal depression may limit the generalizability of findings" is unclear and inconsistent with the information presented in the rest of the protocol. From what I understood women who scored a 9 or above on the SRQ-20 were included. While not a diagnosis, women with depression could be included in that group. More clarity is needed surrounding what this point is trying to say.

Response: We are thankful to the reviewer for her insightful comment!
We agree with the reviewer's comment that women who scored 9 and above on the SRQ could experience depressive symptoms as well. Therefore, we are retracting the statement given i.e. "exclusion of women with perinatal depression may limit the generalizability of findings" from the manuscript (please see page 3, line 75 the manuscript).

Comment 3: It was unclear until the description of the intervention table that the intervention started in pregnancy. It would be helpful to bring this point out earlier and even discuss some of the potential benefits of starting during the antenatal period. One way to do this would be to include some brief information in the introduction about the potential risks of antenatal stress on foetal development and infant cognitive and developmental outcomes (could include work by Prof Vivette Glover for example). I believe starting in pregnancy is a real benefit of the intervention and therefore should be highlighted.

Response: We are thankful to the reviewer for her valuable suggestion!
As suggested, we have mentioned in the introduction section that antenatal stress can have detrimental effects on fetal development (Glover, 2014) (please see page 3, line 88-89 of the manuscript).

We have mentioned in the intervention section that intervention delivery will commence from third trimester of pregnancy and will end at 12-month postpartum (please see page 6, line 235-236 of the manuscript).

Reference:

- Glover, V. (2014). Maternal depression, anxiety and stress during pregnancy and child outcome; what needs to be done. *Best practice & research Clinical obstetrics & gynaecology*, 28(1), 25-35.

Comment 4: The intervention includes 15 sessions across one year which is extensive. It might help to include a short justification for why this intervention duration was chosen. You mention briefly potential issues this might create in terms of scale-up and cost. Therefore, I think more explanation on why this long intervention period was chosen would be beneficial.

Response: We are thankful to the reviewer for her comment!

Our intervention development is informed by an extensive literature review exercise that was led by the WHO (Rahman et al., 2020). In this literature review, we evaluated randomized controlled trials of psychotherapeutic interventions for common maternal mental health problems among women to improve early childhood development in low- and middle-income countries. Since, our aim is to develop a common elements based intervention package to address early infant stimulation, responsive feeding and maternal distress from third trimester of pregnancy till 12-months postpartum, therefore, a 15-months duration of the intervention covers this critical time-period.

Reference:

- Rahman A, Fisher J, Waqas A et al., World Health Organization recommendation on psychotherapeutic interventions for common maternal mental health problems among women to improve early childhood development in low- and middle-income countries: Report of systematic review and meta-analysis of RCTs. Improving Early Childhood Development: WHO guideline. World Health Organization. 2020.

Comment 5: You mention that participants in the TAU group who show signs of severe psychiatric disorders will be referred on to a tertiary mental health facility. What about participants in the intervention group? Will they also be referred on? Generally, the protocol would benefit from more information about other ethical considerations made, especially around disclosure. I know item 17 of the SRQ-20 asks if participants have thought of ending their life. What are the processes in place if women answer yes to this item or disclose that they have thought of or attempted suicide? Also, what if a woman discloses intimate partner violence within the sessions? Some explanation of how these will be managed would be helpful to include.

Response: We are thankful to the reviewer for her comment!

We would like to clarify that participants in both arms who show signs of severe psychiatric disorders that require immediate specialist treatment and follow-up, they will be referred to tertiary mental health care facility, Institute of Psychiatry, Benazir Bhutto Hospital, Rawalpindi.

That was an omission at our part and we have corrected it now (please see page 11, line 284).

Moreover, a detailed section on the ethical considerations has also been added in the manuscript as suggested (see page 14-15, line 426-437 of the manuscript).

Other ethical considerations

In the present study, a referral pathway, based on the existing referral system, has been established to ensure the safety of the study participants. The Institute of Psychiatry (IoP) is the tertiary mental health care facility in the north of Pakistan and situated within the public tertiary healthcare facility and teaching hospital, the Benazir Bhutto Hospital, Rawalpindi. It caters to the mental health needs of the populations in the metropolitan city of Rawalpindi and is the specialist referral facility for common mental health problems identified at the adjacent primary healthcare centers. In our study, if a participant discloses any event of abuse, intimate partner violence, or is at imminent risk of harm to herself and others, identified at the time of assessment or during the intervention delivery, the research team will record the event and referral to the IoP will be made for appropriate medical, medico-legal and psychosocial assessment and rehabilitation.

Comment 6: How will the self-report measures be administered? Will they be administered on paper, through tablets or orally by trained research assistants? This just needs to be made clear.

Response: We are thankful to the reviewer for her comment!

The self-report outcome measures will be orally administered by the trained assessment team blind to the allocation status of the participants (please see page 11, line 293-294).

Comment 7: When you discuss potential mediators you mention maternal distress but do not give a hypothesis or rationale as to why you would investigate this as you do for the other potential mediating variables. It would help with consistency and clarity to include this information.

Response: We are thankful to the reviewer for her comment!

The text has been modified as indicated (please see page 16, line 50-527). The text has been reproduced here as well for ease.

We will explore pathways through which the intervention may influence primary outcome (i.e. mother-child interaction) . There is strong empirical evidence that the quality of the mother-child interactions, such as her sensitivity and responsiveness, is associated with both maternal depression and child development [36-39]. there is also some evidence that perinatal depression interventions, especially ones that include a parenting component, can improve maternal distress and maternal responsiveness, resulting in improved developmental outcomes [40]. We will assess maternal responsiveness with the maternal items of OMCI that measure maternal responsive behaviors during mother-child interaction [41]. The OMCI has been used to evaluate the impact of maternal interventions on maternal responsiveness [44]. Maternal distress will be assessed using SRQ. The tool has been used extensively in the study settings previously to measure maternal distress (Rahman et al., 2005).

Social support; Low social support is a consistent predictor of mental ill health among mothers [45, 46]. Social support can be theorized to be either a mediator or effect modifier; since, building social support is a key feature of the integrated intervention, we hypothesize that increased social support will mediate the relationship between the intervention and child outcomes. Maternal empowerment; Maternal empowerment will be assessed using two items previously used in the same population [32]. Breastfeeding and other health related variables; Perinatal depression interventions similar to this integrated intervention have resulted in higher rates of exclusive breastfeeding at 6 months and increased likelihood of completion of scheduled immunizations [6]. These health-related behaviours may contribute to the positive effect of intervention on improving maternal wellbeing and infant development, we will assess the impact of behaviours related to feeding (breastfeeding, introduction of solid food) and immunizations on mother-child interaction.

Comment 8: For Figure 1, I would suggest including when in pregnancy women will be enrolled and when the baseline measurement is taken (e.g. 3rd trimester).

Response: We are thankful to the reviewer for her comment!

Figure 1 has been edited as indicated.

Reviewer: 3

Dr. Lara Pierce, Boston Children's Hospital

Comment 1: Title "Evaluating the impact of a common elements based intervention to improve the maternal..." "the" should be removed from the title.

Response: We are thankful to the reviewer for her comments!

We have revised the title as indicated (please see 1, line 1-3).

"Evaluating impact of a common elements based intervention to improve maternal psychological wellbeing and mother-infant interaction in rural Pakistan: Study protocol for a randomized controlled trial"

Comment 2: Article Summary- Mixed methods process evaluation is described to assess blended methodology required as a result of COVID-19, however I do not see further discussion of this in the manuscript. It would be helpful for this, as well as other COVID-19 considerations, to be discussed in another section of the article aside from just the summary.

Response: We are thankful to the reviewer for her comment!

We agree with the reviewer's comment and have discussed the aspect of COVID-19 in the other sections of the manuscript as well, while trying to keep the manuscript within the assigned word limit. In the introduction section we have mentioned that delivering such interventions during COVID-19 pandemic becomes very important because COVID-19 is indirectly impacting maternal and child health outcomes due to disruption of health care services and lack of access to existing maternal and child health care during lockdown (Busch-Hallen et al, 2020) (page 3, line 91-94).

In the intervention section, we have mentioned that during the COVID-19 pandemic, when the in-person sessions are not possible, we are delivering the scheduled intervention session via telephone. All those sessions delivered via telephone are not repeated in-person (page 6, line 236-238).

In the outcome measures section, we have mentioned that during COVID-19 pandemic all in-person assessments will be conducted by following the appropriate precautionary measures (page 14, line 406-207).

References:

- Busch-Hallen, J., Walters, D., Rowe, S., Chowdhury, A., & Arabi, M. (2020). Impact of COVID-19 on maternal and child health. *The Lancet Global Health*, 8(10), e1257.

Comment 3: It is noted that exclusion of women with perinatal depression may limit the generalizability of findings, however it is not clear to me when/how women with perinatal depression are excluded. Moreover, the manuscript seems to be framed around addressing perinatal depression as an intervention target, which suggests to me that these women should be included. Could the authors clarify this point?

Response: We are thankful to the reviewer for her insightful comment!

We agree with the reviewer's comment that women who score positive on the SRQ could experience depressive symptoms as well. Therefore, we are retracting the statement given (i.e. exclusion of women with perinatal depression may limit the generalizability of findings) from the manuscript (please see page 3, line 75 of the manuscript).

Comment 4: Introduction- Could the authors please define the common elements approach, as it is a term not all readers will be familiar with. More detail surrounding from where packages of care were chosen, and why, would also be beneficial.

Response: We are thankful to the reviewer for her comment!

A brief description of the common elements approach has been added in the manuscript as suggested (please see page 3, line 106-109 of the manuscript).

“Common Elements Approach (CEA) is the concept of identifying components that overlap across multiple evidence-based practices for a specific problem area [9]. CEA combines a number of overlapping theoretical approaches into a single intervention with synergistic effects across the range of domains.”

Regarding selection of intervention packages, it has been addressed above in an earlier comment. Please see our response to comment 4 (Reviewer 2).

Comment 5: The phrasing “the mother, in turn, requires...via a well monitored and supervised delivery system” suggests to me that mothers need to have very specific training to effectively care for children, which I do not think the authors mean to imply. Would replacing “required” with “receives” retain the meaning the authors intended?

Response: We are thankful to the reviewer for her comment!
We agree with reviewer’s suggestion and have edit the text accordingly (please see page 3, line 110).

Comment 6: Method- Could the authors provide the dates that the intervention did/will take place?

Response: The study dates have been added as indicated (please see page 2, line 57 of the manuscript).

Comment 7: How was the estimate of 20% attrition determined?

Response: We are thankful to the reviewer for his comment!
The attrition rate was determined based on a community based study, conducted in the study settings, where 13% attrition rate was used for 6-months follow-up (Sikander et al, 2019). As the participants in the present study will be followed-up for up to 12-months, the attrition of 20% was considered appropriate.

Comment 7: Given that the SRQ reflects psychological distress across domains (e.g., including depression, anxiety), can the authors clarify whether a score >9 indicates risk of postnatal depression specifically, or whether it represents a more general mental health risk.

Response: We are thankful to the reviewer for her insightful comment!
While SRQ 20 reflects psychological distress across domains, a score of > 9 indicates increased risk of maternal distress/post-partum depression (Rahman et al., 2005; Santos et al., 2007).

Reference:

- Rahman A, Iqbal Z, Lovel H, et al. Screening for postnatal depression in the developing world: a comparison of the WHO Self-Reporting Questionnaire (SRQ-20) and the Edinburgh Postnatal Depression Screen (EPDS). *JPPS* 2005;2:69–72.
- Santos, I. S., Matijasevich, A., Tavares, B. F., da Cruz Lima, A. C., Riegel, R. E., & Lopes, B. C. (2007). Comparing validity of Edinburgh scale and SRQ20 in screening for post-partum depression. *Clinical practice and epidemiology in mental health : CP & EMH*, 3, 18. <https://doi.org/10.1186/1745-0179-3-18>

Comment 8: Are any additional resources provided to mothers who score >9 on the SRQ, regardless of treatment arm? I am wondering whether there is a clinically significant cut-off beyond which it is ethical practice to provide some type of referral or intervention.

Response: We are thankful to the reviewer for her comment!
All trial participants, regardless of their treatment allocation, will have access to the free of cost preventive health care services offered by primary health centers in their community area. Any trial participant, who shows signs of severe psychiatric disorders or severity of symptoms that require immediate specialist treatment and follow-up, will be referred to a tertiary mental health care facility, i.e., Institute of Psychiatry, Benazir Bhutto Hospital, based in the Rawalpindi district. A complete record of services received by trial participants in both arms will be made using the Client Services Receipt Inventory (CSRI) at the end-point follow-up.
This has been addressed above in our earlier response to comment 5 (Reviewer 2). A detailed section on other ethical consideration has also been added in the manuscript (page 14-15, line 428-439).

Comment 9: Can the authors describe how these particular common elements (e.g., WHO, UNICEF, etc.) were chosen for this intervention, as opposed to others?

Response: We are thankful to the reviewer for her comment!

The current study is led by a multidisciplinary team of experts who have vast experience in designing, implementing and evaluating packages of care for improving maternal mental health and child development (Rahman et al 2008; Hamdani et al, 2015; Sikander et al, 2019;). We are working with the WHO on a global need that has been identified by the WHO, UNICEF and its partner countries to develop a core package of care based on common elements approach to promote nurturing care in first 1000 days of life. We have contributed to develop guidelines to provide direction for strengthening policies and programs to better address early childhood development (Rahman et al, 2020). It is through the review of exiting literature and intervention development workshops with the experts that led to the identification of common elements from these packages of care.

We have also addressed this in our earlier response to comment 4 (Reviewer 2).

References:

- Hamdani, S. U., Akhtar, P., Nazir, H., Minhas, F. A., Sikander, S., Wang, D., ... & Rahman, A. (2017). WHO Parents Skills Training (PST) programme for children with developmental disorders and delays delivered by Family Volunteers in rural Pakistan: study protocol for effectiveness implementation hybrid cluster randomized controlled trial. *Global mental health*, 4.
- Sikander, S., Ahmad, I., Atif, N., Zaidi, A., Vanobberghen, F., Weiss, H. A., ... & Rahman, A. (2019). Delivering the Thinking Healthy Programme for perinatal depression through volunteer peers: a cluster randomised controlled trial in Pakistan. *The Lancet Psychiatry*, 6(2), 128-139.
- Rahman, A., Malik, A., Sikander, S., Roberts, C., & Creed, F. (2008). Cognitive behaviour therapy-based intervention by community health workers for mothers with depression and their infants in rural Pakistan: a cluster-randomised controlled trial. *The Lancet*, 372(9642), 902-909.
- Rahman A, Fisher J, Waqas A et al., World Health Organization recommendation on psychotherapeutic interventions for common maternal mental health problems among women to improve early childhood development in low- and middle-income countries: Report of systematic review and meta-analysis of RCTs. *Improving Early Childhood Development: WHO guideline*. World Health Organization. 2020.

Comment 10: Can the authors provide examples for “guided discovery using pictures” and “behavioral activation”?

Response: We are thankful to the reviewer!

Given the word count limit we have not expended on the intervention section, however, the details on these intervention strategies can be accessed in the Thinking Healthy Program manual which is accessible on the WHO website using this link and has been cited in the manuscript. Furthermore, we will make the intervention manual available as an open access resource along with the manuscript describing the results of the study.

Reference link:

- https://www.who.int/mental_health/maternal-child/thinking_healthy/en/

Comment 11: I am curious about the decision to introduce the breastfeeding module at 8 weeks postpartum. I could see benefit to introducing this module at the follow-up after delivery given that that is when the breastfeeding relationship is likely to be established. Can the authors clarify the rationale for including this only at a later point?

Response: We are thankful to the reviewer for her comment and agree that the introduction to breastfeeding need to be made earlier. We have done so in this intervention. Please refer to the table 1, titled "Description of Intervention session" (page 6), the importance of exclusive breastfeeding and colostrum is introduced at 9th month of pregnancy, just before delivery of a baby. Session 5, (Improving child's physical health), introduced at 8-weeks covers the detailed content on exclusive breastfeeding to reinforce this behavior. As, this session is the first main session that is given after birth of a baby, placement of content ensures that mothers can discuss challenges and experience of breastfeeding to a newborn with non-specialist providers in a timely manner.

Comment 12: Can the authors include information about how many LHW visits mothers receive, and the timeline for those visits? Would they typically extend through the first postpartum year (as the intervention does) or are there differences in visit schedule across intervention and TAU groups?

Response: We are thankful to the reviewer for her comment!

Lady Health Workers (LHWs) are based at the Basic Health Units (BHUs) within their respective villages. They are responsible for about 100-150 households in their village catchment areas and keep a record of all women of reproductive age, register every new pregnancy in their catchment area and provide preventive and promotive health care services to all children under five-years of age. LHWs visit new mothers on a monthly basis to monitor the progress of a new-born baby on key parameters such as weight, height, immunization and nutrition. Moreover, LHWs are given additional duties (to provide information about infectious disease or to administer oral polio drops to children) which requires regular visits to houses. In the current study, the number of visits mothers receive from LHWs will be the same for each trial participant, regardless of their allocation status. A complete record of visit received from LHWs by trial participants in both arms will be made using the Client Services Receipt Inventory (CSRI) at the end-point.

Comment 11: Can the authors clarify whether/how baseline differences in mother-child interactions between intervention and TAU groups will be assessed, given that the OMCI is only administered at the end of the intervention?

Response: We are thankful to the reviewer for her comment!

We would like to clarify that the study is powered to make a single comparison in the primary outcome at 12 months only.

Comment 12: Could the authors discuss any validation/norming of the BSID III and ASQ3 that is relevant for cross-cultural administration?

Response: We are thankful to the reviewer for her comment!

The Bayley Scales of Infant and Toddler Development Third Edition (BSID-III) and Ages and Stages Questionnaire (ASQ), including other child outcomes such as HOME and child and growth and development indicators have been extensively used by our research group in similar study settings of Pakistan (Sikander et al, 2019). Moreover, the BSID-III and ASQ3 have been used in community settings of India to measure child development (Kvestad et al, 2013). The required information and relevant references have been cited in the manuscript (please see page 11, line 321-322, page 12, line-327-328 of the manuscript).

References:

- Bhopal, S. S., Roy, R., Verma, D., Kumar, D., Khan, B., Soremekun, S., ... & Kirkwood, B. R. (2021). Using the Mothers Object Relations Scale for early childhood development research in rural India: Findings from the Early Life Stress Sub-study of the SPRING Cluster Randomised Controlled Trial (SPRING-ELS). *Wellcome Open Research*, 6(54), 54.

- Kvestad, I., Taneja, S., Kumar, T., Bhandari, N., Strand, T. A., Hysing, M., & Study Group (2013). The assessment of developmental status using the Ages and Stages questionnaire-3 in nutritional research in north Indian young children. *Nutrition journal*, 12, 50. <https://doi.org/10.1186/1475-2891-12-50>

- Sikander, S., Ahmad, I., Bates, L. M., Gallis, J., Hagaman, A., O'Donnell, K., ... & Maselko, J. (2019). Cohort Profile: Perinatal depression and child socioemotional development; the Bachpan cohort study from rural Pakistan. *BMJ open*, 9(5), e025644.

Comment 13: Under “Early breastfeeding (EBF)” I believe the term “artificial foods” is incorrect as it refers to “milk, cereal, or other food items” which are not artificial.

Response: We are thankful to the reviewer for her comment!

Yes, we agree with the reviewer and have edited the text to indicate it as ‘additional food’ (please see page 12, line 338 of the manuscript).

“This information will be categorized as either EBF, partial breastfeeding (giving an infant some breastfeeding, and some additional foods, either milk, cereal, or other food items), or no breastfeeding.”

Comment 14: The authors state that “missing data will be treated as missing at random,” but can that be determined prior to data collection? How will the authors determine that data are missing at random, and will comparisons between those with and without data be compared on other measures to determine systematic differences between those who did and did not complete a measure?

Response: We are thankful to the reviewer for her comment!

We have revised our Statistical Analysis Plan in consultation with the trial statistician. The revised text incorporates the details on how missing data in the current trial will be treated (please see page 15, line 462-499 of the manuscript). The revised text has been reproduced here as well for ease.

Missing primary outcome and secondary outcomes with a single measurement will not be imputed but missing secondary outcomes with repeated measurements will be imputed in sensitivity analysis using the last observation carried forward strategy. Missing baseline covariates will be imputed using simple imputation methods in the covariate adjusted analysis based on the covariate distributions in the sample. For a continuous variable, missing values will be imputed from random values from a normal distribution with mean and SD calculated from the available sample. For a categorical variable, missing values will be imputed from random values from a uniform distribution with probabilities P_1 , P_2 , ..., and P_k from the sample. Seed for the imputation is set as the date of data analysis (eg, 270421).

We have elaborated the procedure of handling missing data in the manuscript as well. Please see page 16, line 489-499. The test is reproduced here for ease.

Comment 15: Potential mediators: Can the authors clarify why maternal responsiveness is assessed using the HOME inventory as opposed to (or in addition to) being coded from the mother-child interactions? Given that potential mediators and child outcomes are measured at the same time point, can the direction of effects be inferred? For example, if more responsive interactions and higher cognitive scores are observed, could it alternatively be the case that infants with higher cognitive scores elicit more responsive interactions?

Response: We are thankful to the reviewer for her suggestion. We have edited the text in the manuscript to indicate that maternal responsiveness will be measured using maternal items of OMCI instead of HOME inventory (page 16, line 509-514).

We agree with the reviewer that this is an exploratory mediation analysis; therefore, we cannot be certain about the direction of the mediating effect at this stage.

Comment 16: SPIRIT Checklist- “Strategies for achieving adequate participant enrolment to reach target sample size” and “Plans to promote participant retention and complete follow-up, including list of any outcome data to be collected for participants who discontinue or deviate from intervention protocols” are both marked as N/A. However, participant enrolment and retention through a longitudinal study of this nature is critical to its success. Could the authors describe strategies to achieve adequate enrolment and retention?

Response: Thank you for your valuable comment! We have revised the manuscript to reflect the suggested changes. (please see page 4, line 151-159).

The study participants are permanent residents in the study sub-district and have been living here for past many generations. The implementing research organization has been working closely with the health department in the study area for last 15 years and has developed a trustable relationship with the community members. As the study participants in the present study are being approached through Lady Health Workers (LHWs), who have trusting relationship with community, this makes the enrollment process feasible. Also, convenience of participants is taken in account when assessment and intervention delivery are scheduled. Therefore, it is less likely that participants’ follow-up rate will be low in the present study as these factors will improve the retention rate of participants in study over longer period of time.

Comment 17: “Criteria for discontinuing or modifying allocated interventions for a given trial participant (eg, drug dose change in response to harms, participant request, or improving/worsening disease)” is marked N/A. Are there circumstances under which the intervention would be discontinued? (e.g., conditions developed by mother or baby, lack of adherence to protocol, etc.)?

Response: We are thankful to the reviewer for her comment!

It was an omission at our part. The required information is already mentioned in the manuscript on page 5, line 171-176 (exclusion criteria). We have now referred the relevant page no. in the SPIRIT Checklist as well.

VERSION 2 – REVIEW

REVIEWER	Katie Rose Sanfilippo Goldsmiths University of London, Psychology
REVIEW RETURNED	01-Jun-2021

GENERAL COMMENTS	I am happy with the authors' responses to my comments and feel that they have adequately addressed all points. Therefore I suggest the paper is ready for publication. I enjoyed reading this work and look forward to the results of the trial.
--

REVIEWER	Lara Pierce Boston Children's Hospital
REVIEW RETURNED	16-Jun-2021

GENERAL COMMENTS	Thank you to the authors for their consideration of the reviews received. My only remaining comment is a clarification on my part. Regarding the removal of "the" from the title - my intention was just for the second "the" to be removed, not the first. So the title would read: Evaluating the impact of a common elements based intervention to improve maternal psychological wellbeing and mother-infant interaction in rural Pakistan: Study protocol for a randomized controlled trial. My apologies for any confusion!
---

VERSION 2 – AUTHOR RESPONSE

Reviewer: 2

Ms. Katie Rose Sanfilippo, Goldsmiths University of London

Comment 1: I am happy with the authors' responses to my comments and feel that they have adequately addressed all points. Therefore, I suggest the paper is ready for publication. I enjoyed reading this work and look forward to the results of the trial.

Response: We are thankful to the reviewer for her encouraging remarks

Reviewer: 3

Dr. Lara Pierce, Boston Children's Hospital

Comment 1: Thank you to the authors for their consideration of the reviews received. My only remaining comment is a clarification on my part. Regarding the removal of "the" from the title - my intention was just for the second "the" to be removed, not the first. So the title would read: Evaluating the impact of a common elements based intervention to improve maternal psychological wellbeing and mother-infant interaction in rural Pakistan: Study protocol for a randomized controlled trial. My apologies for any confusion!

Response: We are thankful to the reviewer for her comments!

We have revised the title as indicated (please see page 1, line 1-3).

“Evaluating the impact of a common elements based intervention to improve maternal psychological wellbeing and mother-infant interaction in rural Pakistan: Study protocol for a randomized controlled trial”